# Regulatory Effects of Skate Skin-Derived Collagen Peptides with Different Molecular Weights on Lipid Metabolism in the Liver and Adipose Tissue

**DOI:** 10.3390/biomedicines8070187

**Published:** 2020-06-30

**Authors:** Minji Woo, Jeong Sook Noh

**Affiliations:** 1Busan Innovation Institute of Industry, Science & Technology Planning (BISTEP), Busan 48058, Korea; woo07140@pusan.ac.kr; 2Department of Food Science and Nutrition and Kimchi Research Institute, Pusan National University, Busan 46241, Korea; 3Department of Food Science and Nutrition, Tongmyong University, Busan 48520, Korea

**Keywords:** collagen, skate skin, obesity, lipid metabolism, liver, adipose tissue

## Abstract

This study investigated the effects of skate skin collagen peptide (SSCP) with different molecular weights (MWs) on the lipid metabolism in the liver and adipose tissue. Male *db/db* mice were orally administered with water (control group) or low SSCP (LCP group) or high SSCP (HCP group) MW for 8 weeks whereas male *m/m* mice were used for comparison (normal group) (*n* = 10 each group). Compared to the control group, the LCP and HCP groups had lower adipose tissue mass, plasma and hepatic lipid concentrations, and plasma leptin levels (*p* < 0.05). Protein expression levels of lipogenesis-related protein were reduced in both liver and adipose tissues of SSCP-fed groups whereas those for lipolysis were elevated (*p* < 0.05). In particular, the LCP had the higher effects relative to the HCP. The above results were supported by histological analysis, revealing that SSCP administration decreased the size of adipose droplets and suppressed hepatic lipid accumulation. Our results showed that SSCP has potential antiobesity properties through the improvement of lipid metabolism in the liver and adipose tissue; in particular, the lower MW of collagen peptide had the greater effects.

## 1. Introduction

During the processing of fish, byproducts are generated from the skin, bones, fins, heads, and scales, in which the collagen-containing materials account for up to 30% of the total byproducts [1]. Collagen is a major component of animal-derived proteins, and has a wide range of applications, including the cosmetic, pharmaceutical, and food industries, due to its numerous beneficial effects, including antioxidative, lipid lowering, and so on [1,2,3,4,5,6]. A series of studies have verified the beneficial antioxidative effects of marine collagen peptides [2,3], the improvement of glucose/lipid metabolism [2,4,5], and the facilitation of learning and memory function [6]. The bioactivity of collagen is suggested to be directly correlated with its molecular weight (MW) [7]. Some reports suggest that small peptides (composed of 2–10 amino acids) possess more potent therapeutic effects than their parent native proteins or polypeptides (composed of 10–50 amino acids or more) [8]. Antioxidative activities of collagen peptides with a lower MW were greater than those with a higher MW [9]. However, the effect of different MWs of skate skin collagen peptide on lipid metabolism and the related mechanisms of action have not been elucidated.

Obesity is a multifactorial disease that is considered the most challenging public health problem [10,11,12,13]. In obesity, excessive lipid accumulation in the liver and adipose tissue occurs due to the dysregulation of key transcription factors and related enzymes [11,12,13]. At the molecular level, sterol regulatory element-binding protein (SREBP)-1 and -2 stimulate fatty acid and cholesterol synthesis by regulating fatty acid synthase (FAS), acetyl-CoA carboxylase (ACC), and 3-hydroxy-3-methylglutaryl-CoA reductase (HMGCR), respectively [14]. Moreover, adipocyte protein 2 (AP2), a fatty acid-binding protein, regulates the terminal differentiation of adipocytes. In contrast, peroxisome proliferator-activated receptor α (PPARα) upregulates genes involved in fatty acid oxidation, such as carnitine palmitoyltransferase 1A (CPT1) and acyl-CoA oxidase 1 (ACOX1) [15]. In addition, lipolysis, one of the most important processes to reduce fat mass, is regulated by lipases, such as hormone-sensitive lipase (HSL), particularly in triglyceride (TG)-rich adipose tissue [16,17]. These factors coordinately modulate lipid metabolism.

Skate (*Raja kenojei*) is a cartilaginous fish that is currently drawing much attention as a health marine product. Almost all studies on skate skin collagen have focused on the processing conditions [18], purification, and characterization [18,19]. Our previous study demonstrated that skate skin collagen peptide (without ultrafiltration, MW 1050 Da) prevents an increased hepatic lipid concentration in high-fat diet (HFD)-induced obese mice [20] and improves lipid profiles in the plasma and liver of *db/db* mice [21,22]. We hypothesized that collagen peptides with different MWs will have different effects in obese mice. Therefore, the present work evaluated the effect of two skate skin-derived collagen peptides with different MWs on plasma and hepatic lipid profiles, as well as lipogenesis and lipolysis-related markers in the liver and adipose tissue.

## 2. Materials and Methods

### 2.1. Preparation of the Collagen Hydrolysate Sample

Skate skin collagen peptides were hydrolyzed using enzymes, such as alcalase^®^ and protamex^®^ (Novozymes, Bagsvaerd, Denmark), for 2 or 6 h and separated by MW cut-off membranes (1 kDa), LCP (MW < 1 kDa), and HCP (MW > 1 kDa). The average MWs of LCP and HCP were 687 and 1442 Da, respectively. The compositions of both samples were similar. The amino acid constituents were as follows: LCP (aspartic acid 7.11%, hydroxy proline 6.85%, threonine 3.42%, serine 5.71%, glutamic acid 10.78%, proline 9.02%, glycine 22.09%, alanine 7.66%, valine 3.34%, methionine 2.15%, isoleucine 2.45%, leucine 3.67%, tyrosine 0.91%, phenylalanine 2.23%, ammonia 0.74%, lysine 3.22%, histidine 1.26%, arginine 7.05%, and others 0.34%) and HCP (aspartic acid 6.46%, hydroxy proline 6.15%, threonine 3.29%, serine 5.38%, glutamic acid 9.93%, proline 8.28%, glycine 19.97%, alanine 6.92%, valine 2.93%, methionine 1.72%, isoleucine 2.20%, leucine 3.32%, tyrosine 0.82%, phenylalanine 2.02%, ammonia 0.72%, lysine 3.49%, histidine 1.38%, arginine 7.84%, and others 7.23%). All collagen peptide samples were provided by Yeongsan Skate Co., Ltd. (Naju, Jeollanam-do, Korea).

### 2.2. Animal Study

C57BLKS/J *db/db* mice and *m/m* mice (male, 7 weeks old) were purchased from Dooyeol Biotech (Seoul, Korea). *db/db* mice are a leptin-deficient animal model, in which deficiency confers susceptibility to obesity, insulin resistance, and type 2 diabetes mellitus whereas *m/m* mice are widely used as a normal control. Animals were raised under controlled room temperature (23 ± 1 °C) and humidity (50 ± 5%) with a 12/12-h light-dark cycle and fed a standard laboratory pellet chow diet and water ad libitum. After an acclimatization period of 1 week, mice were divided into four groups based on body weight. *db/db* mice were orally administered LCP (LCP group) or HCP (HCP group) at a dose of 200 mg·kg bw^−1^·day^−1^ or water (control group, CON) for 8 weeks, and *m/m* mice were orally administered water as a vehicle (NOR group) using a zonde (*n* = 10 per group). The volume of oral administration was 100 µL. Each sample was prepared by dissolving in water and the concentration for oral administration was based on our previous study [21]. Dietary intake was checked daily and body weight was measured every week. After 8 weeks, the mice were subjected to a 12-h fasting period and sacrificed with CO_2_. Blood was collected by cardiac puncture into heparin tubes and the organs were collected after perfusion with ice-cold phosphate-buffered saline (PBS, 10 mM, pH 7.2). Plasma was separated immediately after blood was drawn and the organs, including liver and adipose tissues (visceral, subcutaneous, and epididymis), were stored at −80 °C until use. The animal study was conducted in accordance with the Guidelines for Animal Experiments approved by the University Institutional Animal Care and Use Committee (Approval number: PNU-2016-1354).

### 2.3. Plasma Lipid, Aminotransferase, and Adipokine Levels

TG (AM157S-K), TC (AM202-K), HDL-C (AM202-K), AST (AM203-K), and ALT (AM101-K) levels were measured using commercially available kits (Asan Pharmaceutical Co., Seoul, Korea). Adipokines, such as leptin (#ADI-900-019A, Enzo Life Sciences AG, Lausen, Switzerland), adiponectin (LF-EK0239; AbFrontier, Seoul, Korea), and FFAs (ab65341; Abcam Inc., Cambridge, MA, USA), were evaluated using commercial kits.

### 2.4. Hepatic Lipid Concentration

The liver tissue was homogenized with 10 volumes (*w*/*v*) of PBS (pH 7.4) using a polytron homogenizer (PT-MR 3100; Kinematica Inc., Lucerne, Switzerland). Hepatic lipids were extracted from liver homogenates using chloroform:methanol (2:1, *v*/*v*) solvent according to a previously described method [23]. Hepatic TG and TC concentrations were evaluated with the same commercially available kit used for the plasma lipid concentration.

### 2.5. Histological Analysis

The liver and adipose tissues were fixed in 4% formalin. Fixed liver and adipose tissue were used in frozen and paraffin blocking, respectively. Frozen-blocked liver tissues were cut (3 μm thick) with a microtome (CM1510S-3; Leica, Wetzlar, Germany) and stained with Oil Red O. Paraffin-blocked adipose tissue was sectioned (3 μm thick) using a microtome (Microm HM 325; Thermo Fisher Scientific, Waltham, MA, USA) and stained with hematoxylin and eosin on a coating glass slide. Slides were observed under a light microscope (×100 magnification; Nikon ECLIPSE Ti; Nikon Corp., Tokyo, Japan).

### 2.6. Western Blot Analysis

The protein expression was measured by the Western blot assay, which was performed as previously described using SDS-PAGE [24]. In brief, liver and epididymis adipose tissue liver tissues were homogenized with lysis buffer containing a protease inhibitor cocktail and the protein concentration was determined using the Bio-Rad protein assay kit (Bio-Rad, Hercules, CA, USA). An equal amount of protein was resolved on 10% SDS-PAGE. The separated proteins were electrophoretically transferred to a nitrocellulose membrane, blocked with 5% (*w*/*v*) skim milk solution, and then incubated with primary and secondary antibodies. Protein expression was visualized by enhanced chemiluminescence, detected using the CAS-400 (Core Bio, Seoul, Korea), and calculated using ImageJ software (National Institutes of Health, Bethesda, MD, USA). Protein expression was normalized to that of α-tubulin or β-actin. The primary antibodies used in this study, α-tubulin (ab52866), beta-actin (ab8227), and FAS (ab22759), were purchased from Abcam Inc. (Cambridge, UK), and AP2 (#3208) was purchased from Cell Signaling Technology (Beverly, MA, USA). The following were purchased from Santa Cruz Biotechnology (Santa Cruz, CA, USA): SREBP-1 (sc-8984), ACCα (sc-26817), PPARα (sc-9000), CPT1 (sc-139482), ACOX1 (sc-98499), SREBP-2 (sc-5603), HMGCR (sc-33827), CYP7A1 (sc-25536), and HSL (sc-74489). The dilution ration of primary antibody was 1:500 to 1:1000. The secondary horseradish peroxidase-conjugated antibodies (all from Abcam Inc.) were rabbit anti-goat IgG H & L (ab6741), donkey anti-rabbit IgG H & L (ab6802), and rabbit anti-mouse IgG H & L (ab6728).

### 2.7. Statistical Analysis

Statistical analysis was performed using SPSS version 23 (SPSS Inc., Chicago, IL, USA). Values are presented as the mean ± standard deviation. Statistical significance of differences was assessed by one-way analysis of variance (ANOVA) followed by Duncan’s multiple-range test at *p* < 0.05.

## 3. Results

### 3.1. Body Weight Gain, Organ Weight, Food Intake, and Aminotransferase Activity

As shown in Figure 1, there were no significant differences in the initial body weight and daily food intake among *db/db* mice (Figure 1a,b). The final body weights were significantly increased in *db/db* mice, relative to *m/m* mice, which were reduced by collagen supplements (Figure 1a, *p* < 0.05). The weight of visceral and subcutaneous adipose tissue increased in *db/db* mice but was reduced in the low-MW collagen peptide (LCP) (by 21.0% and 17.4%) and high-MW collagen peptide (HCP) groups (by 19.6% and 13.5%, respectively, Figure 1e,f, *p* < 0.05). However, those of the liver and epididymis adipose tissue were not significantly different among *db/db* mice (Figure 1c,d). The aminotransferase activities were elevated in *db/db* mice in which aspartic acid transaminase (AST) and alanine transaminase (ALT) activity in the LCP and HCP groups (Figure 1g,h, *p* < 0.05) but activity was not significantly different among *db/db* mice.

### 3.2. Effects of Skate Skin Collagen Peptides on Plasma Lipid, Adiponectin, and Leptin Levels, and Lipid Droplets in Adipose Tissue

Compared to *m/m* mice, the plasma lipid concentration that was elevated in *db/db* mice was reduced by collagen treatment (Figure 2). Compared to the CON group, plasma TG, total cholesterol (TC), and low-density lipoprotein cholesterol (LDL-C) levels were significantly lower in the LCP (29.3%, 35.2%, and 66.5%, respectively) and in the HCP groups (26.8%, 35.2%, and 66.8%, respectively) (Figure 2a–c, *p* < 0.05). Therefore, the atherogenic index was significantly reduced in collagen-fed mice by 45.0% and 46.3% in the LCP and HCP groups, respectively (Figure 2d, *p* < 0.05). Plasma free fatty acid (FFA) levels were reduced in collagen-fed mice, but there was no significant difference among *db/db* mice (Figure 2e). In contrast, plasma high-density lipoprotein cholesterol (HDL-C) levels in collagen-fed mice were lower in the CON group, whereas only the LCP-fed mice were significantly elevated by 41.0% (Figure 2f, *p* < 0.05). Compared to *m/m* mice, plasma adiponectin levels were lower in *db/db* mice (Figure 2g), whereas leptin and insulin levels were higher (Figure 2h,i). However, these levels were reversed by collagen supplementation, in which adiponectin levels in the LCP group were significantly elevated by 52.3% and both leptin and insulin levels in the LCP and HCP groups were diminished in the LCP group (by 33.0% and 44.1%, respectively) and the HCP group (by 49.2% and 46.6%, respectively), relative to the CON group (*p* < 0.05).

As shown in the adipose tissue histological results (Figure 2j), the differentiation of lipid droplets was observed in the CON group, relative to *m/m* mice. In contrast, collagen intake suppressed the enlargement of adipocytes; in particular, the degree of lipid droplet size was smaller in the LCP group than in the HCP group. The histological results were in line with the reduced plasma lipid levels in the collagen-fed groups.

### 3.3. Effects of Skate Skin Collagen Peptides on Hepatic Lipid Levels and the Degree of Lipid Accumulation in Liver Tissue

Hepatic TG and TC levels that were elevated in *db/db* mice were significantly decreased in collagen-fed groups (Figure 3a,b, *p* < 0.05). Compared to the CON group, the hepatic TG level in the LCP group was lower by 28.1% and hepatic TC levels in the LCP and HCP groups were reduced by 10.5% and 11.8%, respectively (*p* < 0.05).

Histological analysis of the liver showed that hepatic lipid deposition was prominent in the CON group, relative to *m/m* mice (Figure 3c), whereas collagen intake suppressed lipid accumulation in the liver. In particular, the degree of hepatic lipid accumulation was less severe in the LCP group than in the HCP group, which was consistent with the decreased hepatic TG and TC levels in the collagen-fed groups.

### 3.4. Protein Expression of Fatty Acid Synthesis in the Liver

Compared to the CON group, protein expression levels of SREBP-1 were significantly lower in the LCP and HCP groups by 42.8% and 41.6%, respectively, which was similar to that in the normal (NOR) group mice (Figure 4, *p* < 0.05). As expected, protein expression levels for FAS were significantly lower in the LCP group by 48.6%, and those for ACCα were significantly reduced in the LCP and HCP groups by 32.5% and 29.2%, respectively (*p* < 0.05).

### 3.5. Protein Expression of Beta-Oxidation in the Liver

Protein expression levels of PPARα were significantly elevated in the LCP group by 163.6%, compared to that in the CON group (Figure 5, *p* < 0.05). Similarly, protein expression levels of CPT1 and ACOX1 in the LCP group were higher by 168.5% and 153.9%, respectively (*p* < 0.05).

### 3.6. Protein Expression of Cholesterol Metabolism in the Liver

Compared to the CON group, the protein expression levels of cholesterol metabolism were significantly different only in the LCP group. The levels of SREBP-2 and its target gene, HMGCR, in the LCP group were significantly lower by 42.7% and 42.1%, respectively (Figure 6, *p* < 0.05). In contrast, protein expression of microsomal cytochrome P450 family 7 subfamily A member 1 (CYP7A1) in the LCP group was significantly elevated by 108.2% (*p* < 0.05).

### 3.7. Protein Expression of Lipogenesis-Related Molecules in Adipose Tissue

Compared to the CON group, protein expression levels involved in lipogenesis were significantly different only in the LCP group (Figure 7). The levels of SREBP-1, FAS, and AP2 in the LCP group were significantly lower by 43.7%, 51.0%, and 49.8%, respectively (*p* < 0.05).

### 3.8. Protein Expression of Lipolysis-Related Molecules in Adipose Tissue

Protein expression levels of PPARα in the LCP and HCP groups were significantly higher by 103.1% and 86.0%, respectively, compared to those in the CON group (Figure 8, *p* < 0.05). Protein expression levels of HSL and ACOX1 were significantly higher (by 110.9% and 120.1%, respectively) only in the LCP group (*p* < 0.05). Although the CPT1 level was elevated by collagen intake, there was no significant difference among *db/db* mice.

## 4. Discussion

Fish collagen peptides are widely used in the food, biomedical, and cosmetic industries because of their good biosafety and excellent bioavailability [25,26]. Skate skin is recognized as a potential source of collagen. Peptide MW is the most important factor that exerts functional properties because MW is highly associated with bioavailability [27,28]. Lower peptide MW is reported to reveal a wide range of biological activities than peptides with higher MW [27]. This study evaluated the effect of two different skate skin collagen peptides of different MWs (high and low) on liver and adipose lipid metabolism in obese mice.

Hyperlipidemia is a characteristic feature in an obese state through excessive fat accumulation in major organs, such as the liver and adipose tissue. In contrast, a reduction in body weight is associated with an improvement in the plasma lipid profile [29,30]. In the current study, the collagen-fed groups had lower plasma TG, TC, and LDL-C levels than the CON group, whereas the plasma HDL-C level was higher in the LCP group. The improvement of plasma lipid profiles in the collagen-fed groups might be associated with elevated adiponectin and reduced leptin levels, accompanied by a reduction in body weight gain and visceral and subcutaneous adipose tissue weights. The reduction in plasma lipid levels and adipose tissue weight was higher in the LCP group than in the HCP group, but a large difference between the LCP and HCP groups was not shown. These results are consistent with previous studies showing that fish collagen peptides decreased serum levels of TC, TG, and LDL-C, and increased serum HDL-C levels [25], as well as reduced body weight [25,29]. Similarly, the collagen derived from salmon [31], gray mullet [5], and skate [22] exert plasma lipid-lowering effects. In addition, chum salmon skin-derived collagen peptide increases adiponectin and decreases leptin levels in rats [32]. Similarly, the results of this study indicate that SSCP could have partial effects on regulation of the leptin level, indicating that both LCP and HCP lowers leptin levels. The preventive effect against hyperlipidemia was also observed in a human study, where marine collagen peptides reduced the levels of TG, FFA, TC, and LDL-C, and increased HDL-C levels [2]. Similarly, a recent clinical study shows that the percentage of body fat and body fat mass were found to be significantly better in the intervention group (2 g of skate skin collagen peptide per day) than those of control subjects [33]. These previous results are in good agreement with those of this study. Moreover, the most abundant amino acids in skate collagen are glycine, alanine, and proline [34]. Glycine supplementation is known to be largely effective in reducing weight in overweight and obese individuals by enhancing white-fat loss [10,35,36]. In addition, glycine elevated adiponectin levels and decreased TG plasma concentrations, resulting in the reduction of the adipose cell size in obese animal models [10,37,38]. These results support the hypolipidemic effects of skate skin collagen peptides rich in glycine.

Lipid metabolism is modulated by several transcription factors that target gene expression. In the present study, the hepatic expression levels of proteins involved in fatty acid synthesis (SREBP-1, ACC, and FAS) and cholesterol synthesis (SREBP-2 and HMGCR) were suppressed in the LCP group, relative to the CON group. However, fatty acid oxidation (PPARα, CPT1, and ACOX1) was enhanced by LCP supplementation. As a result, the LCP-fed groups had lower hepatic TG and TC levels, as evidenced by the histological results, demonstrating that hepatic lipid deposits were suppressed. These results are consistent with a previous study showing that collagen peptide derived from chum salmon skin prevented liver steatosis by upregulating PPARα expression in rats [32]. Our previous study demonstrated the dose-dependent effects of skate skin collagen peptide (without ultrafiltration) on the reduction of the hepatic lipid concentration mediated through the regulation of lipid metabolism-related genes in diet-induced obese mice [20]. Likewise, fish collagen peptides suppress palmitate-induced accumulation of lipid vacuoles in hepatocytes [25]. This phenomenon was confirmed in the current study. These results indicate that skate skin collagen peptide may act by reducing lipid synthesis and cholesterol uptake from the liver and facilitating liver fatty acid oxidation.

Consistent with the present observations in the liver, LCP supplementation significantly decreased lipogenesis and lipolysis in adipose tissue. LCP administration downregulated protein expression levels of fatty acid synthesis-related genes (SREBP-1, FAS, and AP2) and elevated the expression of fatty acid oxidation-related genes (PPARα, HSL, CPT1, and ACOX1) in the adipose tissue, as demonstrated by histological analysis of the adipose tissue showing that the LCP-fed mice had smaller adipocytes. Similarly, fish collagen peptides hydrolyzed from significantly inhibited lipid accumulation suppress adipocyte differentiation [7,21]. In addition, in epididymal adipose tissue of HFD-fed mice, the expression of key regulators of adipocyte differentiation, such as CCAAT-enhancer-binding protein (C/EBPα), PPAR-γ, and AP2 genes, is decreased by fish collagen peptide [25]. These observations suggested that skate skin collagen peptide inhibited adipocyte differentiation through a mechanism involving transcriptional repression of the major regulators.

As expected, the effects of LCP were greater than those of HCP. Our results are in agreement with many other studies that reported that the average collagen MW influences functional properties [39]. Peptides with smaller MW have more favorable bioactivities than peptides with higher MW, thereby enzymatic and chemical modifications have been extensively used to improve the functional properties of peptides [40]. Emerging evidence demonstrates that a lower collagen MW shows greater reducing powers, while a higher collagen MW shows lower effects [9]. In addition, peptides from marine species by-products (blue whiting and brown shrimp) suppress appetite by stimulating cholecystokinin secretion, in which small peptides (<1500 Da) exert a higher effect than larger MW peptides [41]. These results were in line with those of another fish peptide study, in which the peptide with the lowest MW from conger eel showed the highest antioxidative activities among peptides with different MWs [27]. These studies indicate that an increase in the collagen MW reduces functional properties. The higher biological effects of low-MW peptides are associated with increased absorption of smaller peptides that can easily cross the intestinal barrier and react in the body [25,40]. In addition, solubility is also an important factor in determining the functional properties of collagen [39]. The reduction in MW considerably improves the solubility of collagens. Because smaller peptides are more polar and form stronger hydrogen bonds with water, these peptides are more soluble [42,43]. Consistent with previous findings, our results showed that the lower collagen peptide MW had a larger effect on the improvement of lipid metabolism in the liver and adipose tissue.

## 5. Conclusions

To identify the potential antiobesity properties of collagen peptide, we report here that skate skin collagen peptide improved lipid metabolism in the liver and adipose tissue. These effects were mediated through the regulation of key transcription factors and enzymes related to lipogenesis and lipolysis. In particular, the low collagen peptide MW augmented larger effects relative to the high collagen peptide MW. Our results are important to broaden the applications of skate skin collagen peptides. Furthermore, advanced studies for metabolic changes by collagen peptide or amino acids are needed to uncover the effects of skate skin collagen peptide.

## Figures and Tables

**Figure 1 biomedicines-08-00187-f001:**
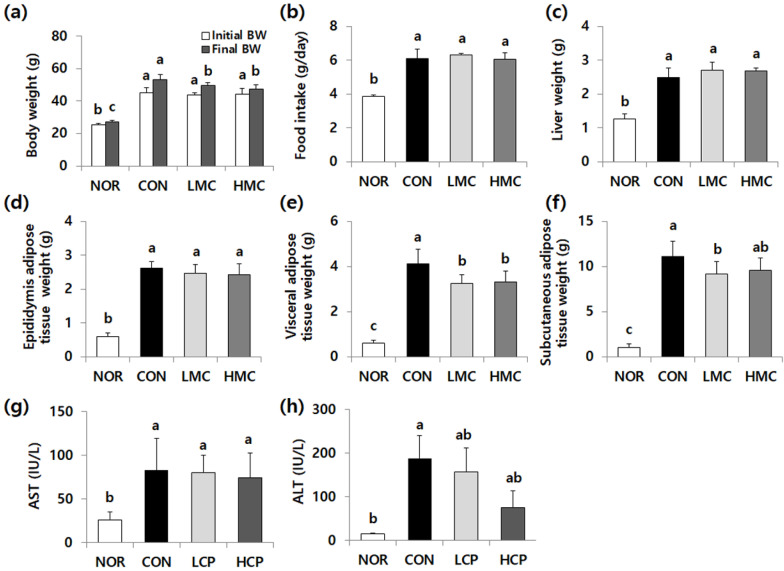
Effects of two skate skin-derived collagen peptides with different molecular weights on body weight (**a**), food intake (**b**), organ weight (**c**–**f**), and aminotransferase activity (**g**,**h**). Data are mean ± standard deviation (SD) (*n* = 10 per group). Experimental groups were divided into four groups; *m/m* mice were orally administered water as a vehicle (NOR group) whereas *db/db* mice were orally administered water (control group, CON), LCP (LCP group), or HCP (HCP group) at a dose of 200 mg·kg bw^−1^·day^−1^ for 8 weeks. ^a–c^ Different letters mean significant differences to one-way analysis of variance (ANOVA), followed by Duncan’s multiple-range test at *p* < 0.05.

**Figure 2 biomedicines-08-00187-f002:**
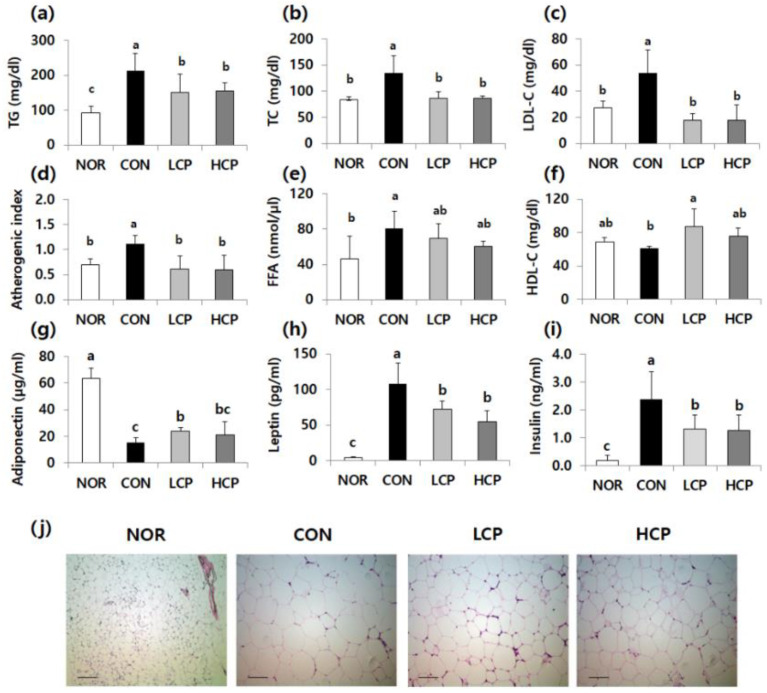
Effects of two skate skin-derived collagen peptides with different molecular weights on plasma lipid (**a**–**f**), adiponectin (**g**), leptin (**h**), and insulin levels (**i**), and lipid droplets in adipose tissue (**j**). Data are mean ± standard deviation (SD) (*n* = 10 per group). Experimental groups were divided into four groups; *m/m* mice were orally administered water as a vehicle (NOR group) whereas *db/db* mice were orally administered water (control group, CON), LCP (LCP group), or HCP (HCP group) at a dose of 200 mg·kg bw^−1^·day^−1^ for 8 weeks. ^a–c^ Different letters mean significant differences to one-way analysis of variance (ANOVA), followed by Duncan’s multiple-range test at *p* < 0.05. The atherogenic index (AI) was calculated as (TC - HDL-C) / HDL-C. Histological analysis of adipose tissue was performed using hematoxylin and eosin staining, magnification: 200×, bar: 50 μm.

**Figure 3 biomedicines-08-00187-f003:**
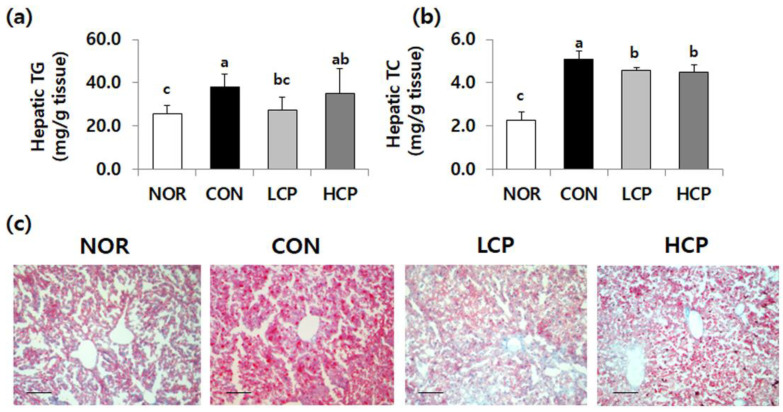
Effects of two skate skin-derived collagen peptides with different molecular weights on hepatic lipid levels (**a**,**b**) and lipid accumulation in the liver (**c**). Data are mean ± standard deviation (SD) (*n* = 10 per group). Experimental groups were divided into four groups; *m/m* mice were orally administered water as a vehicle (NOR group) whereas *db/db* mice were orally administered water (control group, CON), LCP (LCP group), or HCP (HCP group) at a dose of 200 mg·kg bw^-1^·day^-1^ for 8 weeks. ^a–c^ Different letters mean significant differences to one-way analysis of variance (ANOVA), followed by Duncan’s multiple-range test at *p* < 0.05. Histological analysis of liver tissue was performed using oil red O staining, magnification: 100×, bar: 100 μm.

**Figure 4 biomedicines-08-00187-f004:**
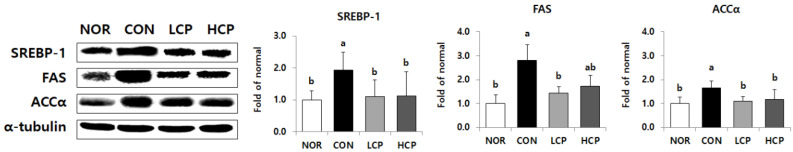
Effects of two skate skin-derived collagen peptides with different molecular weights on hepatic protein expression of fatty acid synthesis. Data are mean ± standard deviation (SD) (*n* = 10 per group). Experimental groups were divided into four groups; *m/m* mice were orally administered water as a vehicle (NOR group) whereas *db/db* mice were orally administered water (control group, CON), LCP (LCP group), or HCP (HCP group) at a dose of 200 mg·kg bw^−1^·day^−1^ for 8 weeks. ^a,b^ Different letters mean significant differences to one-way analysis of variance (ANOVA), followed by Duncan’s multiple-range test at *p* < 0.05.

**Figure 5 biomedicines-08-00187-f005:**
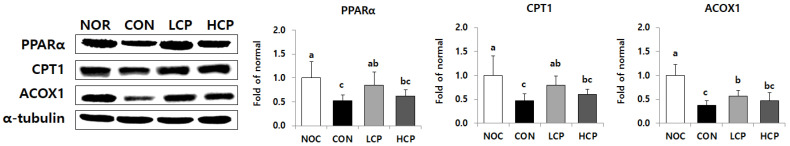
Effects of two skate skin-derived collagen peptides with different molecular weights on hepatic protein expression of beta-oxidation. Data are mean ± standard deviation (SD) (*n* = 10 per group). Experimental groups were divided into four groups; *m/m* mice were orally administered water as a vehicle (NOR group) whereas *db/db* mice were orally administered water (control group, CON), LCP (LCP group), or HCP (HCP group) at a dose of 200 mg·kg bw^−1^·day^−1^ for 8 weeks. ^a-–c^ Different letters mean significant differences to one-way analysis of variance (ANOVA), followed by Duncan’s multiple-range test at *p* < 0.05.

**Figure 6 biomedicines-08-00187-f006:**
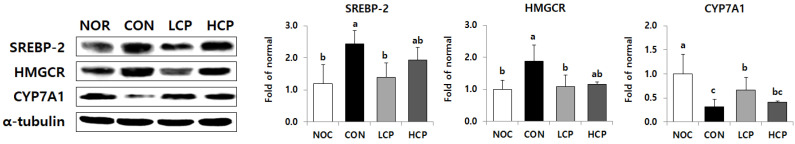
Effects of two skate skin-derived collagen peptides with different molecular weights on hepatic protein expression of cholesterol metabolism. Data are mean ± standard deviation (SD) (*n* = 10 per group). Experimental groups were divided into four groups; *m/m* mice were orally administered water as a vehicle (NOR group) whereas *db/db* mice were orally administered water (control group, CON), LCP (LCP group), or HCP (HCP group) at a dose of 200 mg·kg bw^−1^·day^−1^ for 8 weeks. ^a–-c^ Different letters mean significant differences to one-way analysis of variance (ANOVA), followed by Duncan’s multiple-range test at *p* < 0.05.

**Figure 7 biomedicines-08-00187-f007:**
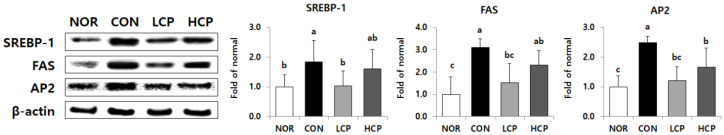
Effects of two skate skin-derived collagen peptides with different molecular weights on protein expression of lipogenesis-related molecules in adipose tissue. Data are mean ± standard deviation (SD) (*n* = 10 per group). Experimental groups were divided into four groups; *m/m* mice were orally administered water as a vehicle (NOR group) whereas *db/db* mice were orally administered water (control group, CON), LCP (LCP group), or HCP (HCP group) at a dose of 200 mg·kg bw^−1^·day^−1^ for 8 weeks. ^a–c^ Different letters mean significant differences to one-way analysis of variance (ANOVA), followed by Duncan’s multiple-range test at *p* < 0.05.

**Figure 8 biomedicines-08-00187-f008:**
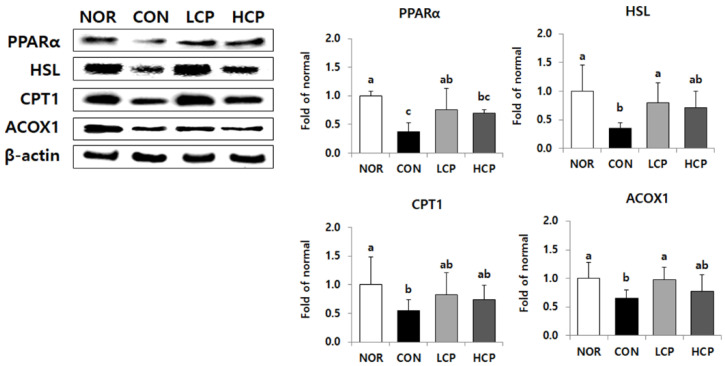
Effects of two skate skin-derived collagen peptides with different molecular weights on protein expression of lipolysis-related molecules in adipose tissue. Data are mean ± standard deviation (SD) (*n* = 10 per group). Experimental groups were divided into four groups; *m/m* mice were orally administered water as a vehicle (NOR group) whereas *db/db* mice were orally administered water (control group, CON), LCP (LCP group), or HCP (HCP group) at a dose of 200 mg·kg bw^−1^·day^−1^ for 8 weeks. ^a–c^ Different letters mean significant differences to one-way analysis of variance (ANOVA), followed by Duncan’s multiple-range test at *p* < 0.05.

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
