# Peer review of "Regulatory Effects of Skate Skin-Derived Collagen Peptides with Different Molecular Weights on Lipid Metabolism in the Liver and Adipose Tissue"

_biomedicines, 2020, doi:10.3390/biomedicines8070187_

Round 1
Reviewer 1 Report
In this manuscript, Woo M. and Noh J.S. describe the anti-obesogenic effects of two different collagen peptides, one with lower molecular weight (LCP) and another with higher molecular weight (HCP). They found that db/db mice that are administered with these collagen peptides had a minor decrease in body weight, lower leptin levels, as well as plasma and hepatic lipid concentrations. They discovered that among these two peptide groups, compared to HCP, LCP is more effective in maintaining lipid homeostasis in the liver.
Major comment:
- The previous publications by these authors already found that skate skin collagen peptides have anti-obesogenic effects and regulate hepatic lipid homeostasis (references 20 and 21 published in 2018). How are these findings more novel or even different compared to these published results? Are the authors trying the compare different molecular weight collagen peptides? If that is the case, this was not reflected neither in the title, nor in the abstract/main text. In addition, the only significant but still minor difference in the effect of these peptides was on hepatic lipid levels and adipose-tissue SREBP levels. I think the manuscript needs a major textual/content revision on clarifying these novelty issues.
Less major comments:
- Although the db/db mice treated with the collagen peptides have lower leptin levels, their food intake rate was unchanged. What could be the cause for this contradiction?
- Glycine is the major component of both groups of collagen peptides. How do the metabolic changes in db/db mice administered LCP/HCP compare to changes associated with glycine supplementation? Glycine is likely the effective component of these peptides and its impact on lipid metabolism is already known.
Author Response
Dear Editor
We have revised our manuscript very carefully based on reviewer’s comments. We are grateful to the critical comments that have helped us to improve the manuscript considerably. The revised parts are highlighted using a red font in the manuscript
Manuscript ID: biomedicines-806246
“Regulatory effects of collagen peptide derived from skate skin on lipid metabolism in the liver and adipose tissue” submitted by Minji Woo and Jeong Sook Noh.
Referee 1
In this manuscript, Woo M. and Noh J.S. describe the anti-obesogenic effects of two different collagen peptides, one with lower molecular weight (LCP) and another with higher molecular weight (HCP). They found that db/db mice that are administered with these collagen peptides had a minor decrease in body weight, lower leptin levels, as well as plasma and hepatic lipid concentrations. They discovered that among these two peptide groups, compared to HCP, LCP is more effective in maintaining lipid homeostasis in the liver.
Major comment:
- The previous publications by these authors already found that skate skin collagen peptides have anti-obesogenic effects and regulate hepatic lipid homeostasis (references 20 and 21 published in 2018). How are these findings more novel or even different compared to these published results? Are the authors trying the compare different molecular weight collagen peptides? If that is the case, this was not reflected neither in the title, nor in the abstract/main text. In addition, the only significant but still minor difference in the effect of these peptides was on hepatic lipid levels and adipose-tissue SREBP levels. I think the manuscript needs a major textual/content revision on clarifying these novelty issues.
A: As reviewer mentioned, author would like to investigate to compare different molecular weight collagen peptides. Our previous publication has been studied on its dose-dependent effects using only one kind of collagen peptide (Ref. 20) or only one dose (Ref. 21). In addition, anti-obesity study has performed in only liver tissue of diet-induced obesity model (Ref 20). However, in this current study, db/db mice were used as well as both liver and adipose tissues were investigated. As reviewer commented, author revised the title and the manuscript (p 2, line 57-63).
Less major comments:
- Although the db/db mice treated with the collagen peptides have lower leptin levels, their food intake rate was unchanged. What could be the cause for this contradiction?
A: These results indicate that collagen peptide could have effects on regulation of leptin level although food intake is similar. Although db/db mice is leptin-deficient animal model, the regulatory effects by collagen could have partially impact on leptin levels. Authors added the above content in the discussion (p 7, line 224-225).
- Glycine is the major component of both groups of collagen peptides. How do the metabolic changes in db/db mice administered LCP/HCP compare to changes associated with glycine supplementation? Glycine is likely the effective component of these peptides and its impact on lipid metabolism is already known.
A: As reviewer mentioned, the effects of glycine on lipid metabolism have been well-known and authors expect the glycine in SSCP could partially effects on lipid metabolism. Author provided the content rate of glycine in LCP (22.09%) and HCP (19.97%) in the method (p 8, line 288, 291) but unfortunately authors could not confirm the effects of metabolic changes by LCP/HCP. Futher study for metabolic changes by collagen peptide or amino acid would be required and author addressed it in the manuscript (p 10, line 363-364)
We hope that the revised manuscript is now acceptable for publication in the ‘Biomedicines’. Thank you very much for your consideration.
Reviewer 2 Report
The authors extend their previous work on skate skin collagen peptides (SSCP) by analyzing the effect of oral administration of low and high molecular weight SSCP in db/db mice compared to vehicle treated animals and to m/m control vehicle treated animals, with a reasonable number of 10 per group. The study is well designed and carefully performed with the analysis of a number of parameters including body and adipose tissue mass, plasma and liver concentrations of lipids and related enzymes, histology as well as protein levels of lipogenesis and lipolysis related proteins. The data provide consistent results on a clear anti-obesity effect of SSCP which is stronger for low molecular weight peptides which for some parameters even lead to normalization.
The manuscript is well structured and written, clear figures visualize and support the results and are carefully prepared. Even though the results are not surprising and the general effect has similarly been described as mentioned by the authors in the discussion, the details and data are convincing and are well prepared. There is only a number of minor issues for some more details and minor corrections as listed below.
Abstract
L15… Mention sex of mice
L17 “as vehicle” is a rather unusual in this context (as also the CON received water as a vehicle), rephrase like “used for comparison” or similar or rephrase to two control groups with vehicle only, db/db and m/m.
L19 serum? Leptin levels
Introduction
L33 As the previous sentence mentions cosmetic and food industries, the type of application in the next sentence should be included, i.t. antioxidative effects/metabolism/learning etc. improved after topical/oral/… application?
L39/40 please check the reference, [9] does not seem to contain information on antioxidative effects as mentioned by the authors.
L43 please check ref [10] – it does not seem to be a general one to support the sentence.
It would be helpful to introduce very briefly the db/db mouse model (including m/m) used in this study as this is a general Biomedicine Journal not only focused on obesity.
Methods
L292 please describe the oral administration (dilution of LCP/HCP, volume range and way of application, single or several doses, during light or dark cycle, handling etc.)
L296 – how was blood obtained?
L314 - which kind of adipose tissue was used`?
Part 4.6. – It would be helpful to provide the antibody dilutions used.
Please provide formula for atherogenic index calculation.
Wester Blot: how were tissues lysed/homogenized (buffer?), protein quantified, how were band intensities compared/standardized (to house keeping proteins, to total protein?) Some figures show a-tubulin, some b-actin.
Results
Please provide exact p-values throughout the text or some other more precise means of significance (e.g. L67 reduction of body weight by LCP - % and p values compared to CON, same for HCP etc., also 4 p-values for L 69/70 etc..)
This would also be helpful as for AST, CON and LCP do not look that different but are apparently different according to the authors, whereas for ALT, CON and HCP look different but are not (significantly) according to the authors.
Figure 1 body weight instead of bodyweight
It could be helpful to label individual graphs of one figure with e.g. letters A, B … to better refer to them (similar for individual figures in 2A etc.)
L85+94 + check throughout: m/m in Italics
L88 “HCP groups (26.8%, 35.2%, and 22.3%, respectively)” – the last number (22,3%) seems to be different from the graph, the bar is similar to LCP and HCP groups for LDL-C and LCP has 66,5% reduction. Please check.
L91f: “In contrast, plasma high-density lipoprotein cholesterol (HDL-C) levels in collagen-fed mice were higher in the CON group,..” – higher compared to which ones? They seem to be lower than in m/m mice.
L96 “elevated by 152.3%” elevated TO instead of BY?
L147 similarly “To” instead of “by” 163% Please check throughout the text for the correct use of to xy% and by xy% (seems to be incorrect for values above 100%)
2.7 and 2.8 – what kind of adipose tissue was used (visceral, subcutaneous, epididymis)?
Discussion
L213f “The reduction in plasma lipid levels and adipose tissue weight was much higher in the LCP group than in the HCP group.” – In figures 1+2 I do not see such a large difference between LCP and HCP (only in later parts of the results it is really clear).
L215 “peptides exhibit .. serum levels” – not a suitable word “exhibit”
Are there any speculations on the mechanisms, how do collagen peptides or abundant amino acids lead to these effects?
Can you please comment the applied dose (200mg/kg bw per day) – is this a dose that would be easily reachable in humans by food supplementation? How does it compare to amounts that would need to be consumed? Do you expect a linear effect – present but lower effect with less amount consumed?
Are there any side effects (known or observed in mice)? Which dose would still be safe to be applied (in mice, in humans)?
How can it be assured that differences in effect between HCP and LCP are not mostly due to solubility problems (as also mentioned in L267) during application?
Author Response
Dear Editor
We have revised our manuscript very carefully based on reviewer’s comments. We are grateful to the critical comments that have helped us to improve the manuscript considerably. The revised parts are highlighted using a red font in the manuscript
Manuscript ID: biomedicines-806246
“Regulatory effects of collagen peptide derived from skate skin on lipid metabolism in the liver and adipose tissue” submitted by Minji Woo and Jeong Sook Noh.
Reviewer 2
The authors extend their previous work on skate skin collagen peptides (SSCP) by analyzing the effect of oral administration of low and high molecular weight SSCP in db/db mice compared to vehicle treated animals and to m/m control vehicle treated animals, with a reasonable number of 10 per group. The study is well designed and carefully performed with the analysis of a number of parameters including body and adipose tissue mass, plasma and liver concentrations of lipids and related enzymes, histology as well as protein levels of lipogenesis and lipolysis related proteins. The data provide consistent results on a clear anti-obesity effect of SSCP which is stronger for low molecular weight peptides which for some parameters even lead to normalization.
The manuscript is well structured and written, clear figures visualize and support the results and are carefully prepared. Even though the results are not surprising and the general effect has similarly been described as mentioned by the authors in the discussion, the details and data are convincing and are well prepared. There is only a number of minor issues for some more details and minor corrections as listed below.
Abstract
L15… Mention sex of mice
A: As the reviewer mentioned, authors added the sex of mice (p 1, line 15, 17).
L17 “as vehicle” is a rather unusual in this context (as also the CON received water as a vehicle), rephrase like “used for comparison” or similar or rephrase to two control groups with vehicle only, db/db and m/m.
A: As the reviewer recommended, authors revised the sentence (p 1, line 17).
L19 serum? Leptin levels
A: As the reviewer recommended, authors added the word “plasma” (p 1, line 19).
Introduction
L33 As the previous sentence mentions cosmetic and food industries, the type of application in the next sentence should be included, i.t. antioxidative effects/metabolism/learning etc. improved after topical/oral/… application?
A: As the reviewer recommended, authors revised the sentence (p 1, line 33-34).
L39/40 please check the reference, [9] does not seem to contain information on antioxidative effects as mentioned by the authors.
A: As the reviewer recommended, authors replaced the reference (p 1, line 40).
L43 please check ref [10] – it does not seem to be a general one to support the sentence.
A: As the reviewer recommended, authors revised the references (p 1, line 44).
It would be helpful to introduce very briefly the db/db mouse model (including m/m) used in this study as this is a general Biomedicine Journal not only focused on obesity.
A: As the reviewer recommended, authors added the brief explanation for db/db and m/m mouse model (p 8, line 297-299).
Methods
L292 please describe the oral administration (dilution of LCP/HCP, volume range and way of application, single or several doses, during light or dark cycle, handling etc.)
A: Oral administration was performed using a zonde and its volume was 100 ul. The sample was prepared by dissolving in water. As the reviewer recommended, authors added the above sentence in the method (p 8-9, line 304-305).
L296 – how was blood obtained?
A: Blood was collected by cardiac puncture into heparin tubes and plasma was separated immediately after blood drawn. As the reviewer recommended, authors added the above sentence in the method (p 9, line 308-311).
L314 - which kind of adipose tissue was used`?
A: Three part of adipose tissues including visceral, subcutaneous, and epididymis were collected and weighed. Among them, epididymis adipose tissue was used in western blot assay. As the reviewer commented, authors added the above sentence in the method (p 9, line 309-311 & line 335-339).
Part 4.6. – It would be helpful to provide the antibody dilutions used.
A: Author provided the dilution ratio of antibody in the method briefly (p 9, line 349-350).
Please provide formula for atherogenic index calculation.
A: Author provided the atherogenic index calculation in the legend of Figure 2.
Wester Blot: how were tissues lysed/homogenized (buffer?), protein quantified, how were band intensities compared/standardized (to house keeping proteins, to total protein?) Some figures show a-tubulin, some b-actin.
A: The lysis buffer was used for homogenization. The protein concentration was determined using the Bio-Rad protein assay kit (Bio-Rad, California, USA). An equal amount of proteins was resolved on 10% SDS-PAGE. The band density was calculated using Image J software and normalized to that of α-tubulin or β-actin. Author added the explanation in the methods (p 9, line 335-344).
Results
Please provide exact p-values throughout the text or some other more precise means of significance (e.g. L67 reduction of body weight by LCP - % and p values compared to CON, same for HCP etc., also 4 p-values for L 69/70 etc..)
A: Statistical analysis was performed using SPSS. Statistical significance of differences was assessed by one-way analysis of variance (ANOVA) followed by Duncan’s multiple-range test at p < 0.05. The result of statistical analysis on ANOVA Duncan’s multiple-range test provides whether there is significant difference among experimental groups and a subset for grouping. Numerous animal studies using ANOVA Duncan’s multiple-range test indicate the statistical significance as p<0.05 or p<0.01.
This would also be helpful as for AST, CON and LCP do not look that different but are apparently different according to the authors, whereas for ALT, CON and HCP look different but are not (significantly) according to the authors.
A: Author rechecked the data and revise the graph for AST in the Figure 1. However, the graph for ALT has been already correct (significance: NOR, b; CON, a; LCP, ab; HCP, ab).
Figure 1 body weight instead of bodyweight
A: As the reviewer recommended, authors revised the word ‘body weight’ in the Figure 1.
It could be helpful to label individual graphs of one figure with e.g. letters A, B … to better refer to them (similar for individual figures in 2A etc.)
A: As the reviewer recommended, authors added the letter in the Figure 1 & 2.
L85+94 + check throughout: m/m in Italics
A: As the reviewer recommended, authors revised the letter in Italics in paragraph 2.2.
L88 “HCP groups (26.8%, 35.2%, and 22.3%, respectively)” – the last number (22,3%) seems to be different from the graph, the bar is similar to LCP and HCP groups for LDL-C and LCP has 66,5% reduction. Please check.
A: As the reviewer recommended, authors rechecked and revised the percentage (p 3, line 89).
L91f: “In contrast, plasma high-density lipoprotein cholesterol (HDL-C) levels in collagen-fed mice were higher in the CON group,..” – higher compared to which ones? They seem to be lower than in m/m mice.
A: As the reviewer commented, authors revised the sentence (p 3, line 94).
L96 “elevated by 152.3%” elevated TO instead of BY?
L147 similarly “To” instead of “by” 163% Please check throughout the text for the correct use of to xy% and by xy% (seems to be incorrect for values above 100%)
A: As authors know, both ‘by’ and ‘to’ are widely used in the other articles. In addition, this manuscript has been received the English editing service by the professional foreigners.
2.7 and 2.8 – what kind of adipose tissue was used (visceral, subcutaneous, epididymis)?
A: The epididymis adipose tissue was used in western blot assay. As the reviewer commented, authors added the explanation in the method (p 9, line 335-339).
Discussion
L213f “The reduction in plasma lipid levels and adipose tissue weight was much higher in the LCP group than in the HCP group.” – In figures 1+2 I do not see such a large difference between LCP and HCP (only in later parts of the results it is really clear).
A: As the reviewer commented, authors revised the sentence (p 7, line 215-219).
L215 “peptides exhibit .. serum levels” – not a suitable word “exhibit”
A: As the reviewer commented, authors revised the sentence (p 7, line 220).
Are there any speculations on the mechanisms, how do collagen peptides or abundant amino acids lead to these effects?
A: Unfortunately, authors did not investigate which or how amino acid in collage peptide can lead to these effects. Instead, amino acid in collagen peptide might have effects on these effects by citing previous study in the discussion.
Can you please comment the applied dose (200mg/kg bw per day) – is this a dose that would be easily reachable in humans by food supplementation? How does it compare to amounts that would need to be consumed? Do you expect a linear effect – present but lower effect with less amount consumed?
Are there any side effects (known or observed in mice)? Which dose would still be safe to be applied (in mice, in humans)?
A: Recently, a clinical study shows that the percentage of body fat and body fat mass (kg) in intervention group (2 g of skate skin collagen peptide (SCP) per day) were found to be significantly better than those of control subjects. In this previous study, no notable adverse effect by SCP was reported. Authors added the sentence in the discussion (p 7, line 227-230).
How can it be assured that differences in effect between HCP and LCP are not mostly due to solubility problems (as also mentioned in L267) during application?
A: As the reviewer’s comment, author did not figure out the accurate solubility of HCP and LCP. However, the slight difference between HCP and LCP might be related to the molecular weight according to numerous previous studies [9, 25, 36, 39 40].
We hope that the revised manuscript is now acceptable for publication in the ‘Biomedicines’. Thank you very much for your consideration.
Reviewer 3 Report
In the current work by Woo and Noh, the authors seek to expand previous studies by testing the effect of skate skin collagen peptide (SSCP), of either low or high molecular weight, on hepatic and adipose lipid metabolism in db/db mice. Collectively, treatment with SSCPs led to improvements in plasma lipids, decreased adipocyte size, and decreased hepatic lipids, which was to a greater extent in the low molecular weight SSCP-treated mice. Lastly, the expression of proteins involved in lipogenesis were decreased while expression of fatty acid oxidation proteins was increased in both adipose and liver. Overall, this work supports the conclusion that low molecular weight SSCP is more efficacious than high molecular weight SSCP, however, there are some issues that should be addressed to improve the solidarity of these studies.
Concerns:
-The AST levels in Figure 1 appear to be almost identical, with a sizeable standard deviation. Are these values really significant?
-Quantification of adipose tissue adipocyte size should be performed to support conclusion of decreased droplet size. Also, please include scale bars in the histology figures.
-Not sure what the relevance is of altered leptin levels in db/db (leptin receptor deficient) mice. Is this just being used as an indicator of decreased fat mass in treatment groups?
-It would be helpful to include any data on levels of glucose and insulin in the blood since there is a strong correlation between obesity, nonalcoholic liver disease, and diabetes.
-What is difference between visceral and epididymal fat?
-It is unclear what the mechanism is for the low molecular weight SSCP, particularly in regards to greater effectiveness than high molecular weight SSCP. Some statements in discussion refer to collagen as a source of peptides, but it was not clearly stated. Is the mechanism that low molecular weight collagen is easier to digest/breakdown, and serves as a source of glycine? Are glycine levels increased in low vs high molecular weight?
-The introduction might be more effective if the obesity problem was stated first, then description of the use of collagen as a treatment (rather than beginning with a description of collagen).
Author Response
Dear Editor
We have revised our manuscript very carefully based on reviewer’s comments. We are grateful to the critical comments that have helped us to improve the manuscript considerably. The revised parts are highlighted using a red font in the manuscript
Manuscript ID: biomedicines-806246
“Regulatory effects of collagen peptide derived from skate skin on lipid metabolism in the liver and adipose tissue” submitted by Minji Woo and Jeong Sook Noh.
Reviewer 3
In the current work by Woo and Noh, the authors seek to expand previous studies by testing the effect of skate skin collagen peptide (SSCP), of either low or high molecular weight, on hepatic and adipose lipid metabolism in db/db mice. Collectively, treatment with SSCPs led to improvements in plasma lipids, decreased adipocyte size, and decreased hepatic lipids, which was to a greater extent in the low molecular weight SSCP-treated mice. Lastly, the expression of proteins involved in lipogenesis were decreased while expression of fatty acid oxidation proteins was increased in both adipose and liver. Overall, this work supports the conclusion that low molecular weight SSCP is more efficacious than high molecular weight SSCP, however, there are some issues that should be addressed to improve the solidarity of these studies.
Concerns:
-The AST levels in Figure 1 appear to be almost identical, with a sizeable standard deviation. Are these values really significant?
A: Author rechecked the data and revise the graph for AST (significance: NOR, b; CON, a; LCP, a; HCP, a) in the Figure 1.
-Quantification of adipose tissue adipocyte size should be performed to support conclusion of decreased droplet size. Also, please include scale bars in the histology figures.
A: As the reviewer’ recommendation, author added the scale bar in the figures.
-Not sure what the relevance is of altered leptin levels in db/db (leptin receptor deficient) mice. Is this just being used as an indicator of decreased fat mass in treatment groups?
A: As the reviewer commented, authors think that the reduced leptin level by collagen peptide might be related to the decreased fat mass. Author discussed about it in the discussion (p 7, line 215-219)
-It would be helpful to include any data on levels of glucose and insulin in the blood since there is a strong correlation between obesity, nonalcoholic liver disease, and diabetes.
A: As the reviewer commented, authors provided plasma insulin level in the Figure 2.
-What is difference between visceral and epididymal fat?
A: Visceral adipose tissue is located deep in the abdomen and around internal organs. Epididymal adipose tissue is a fat attached on testis. It is easy to distinguish between two kinds of adipose tissues.
-It is unclear what the mechanism is for the low molecular weight SSCP, particularly in regards to greater effectiveness than high molecular weight SSCP. Some statements in discussion refer to collagen as a source of peptides, but it was not clearly stated. Is the mechanism that low molecular weight collagen is easier to digest/breakdown, and serves as a source of glycine? Are glycine levels increased in low vs high molecular weight?
A: The glycine content in LCP and HCP used in this study were 22.09% and 19.97%, respectively (p 8, line 288, 291). As the reviewer mentioned, the low molecular weight of collagen can be easy to digest/breakdown and dissolve. Numerous studies indicated that studies indicated that an increase in the collagen MW reduced functional properties. In addition, the higher biological effects of low MW peptides are associated with increased absorption of smaller peptides that can easily cross the intestinal barrier and react in the body. It might be related that smaller peptides are more polar and form stronger hydrogen bonds with water, these peptides are more soluble. Authors discussed it in the manuscript (p 8, line 273-278).
-The introduction might be more effective if the obesity problem was stated first, then description of the use of collagen as a treatment (rather than beginning with a description of collagen).
A: Lots of studies have researched the anti-obesity effects and the description of obesity have well-known. In this study, authors would like to emphasis on the effects of two collagen peptide with different molecular weight.
We hope that the revised manuscript is now acceptable for publication in the ‘Biomedicines’. Thank you very much for your consideration.
Round 2
Reviewer 1 Report
The authors addressed all my questions and added the comments to the discussion part of their manuscript. It can be accepted to publication in this current form.
Author Response
Dear Editor and reviewer
We are grateful to the critical comments that have helped us to improve the manuscript considerably.
Thank you very much for your consideration.

Reviewer 2 Report
The authors have addressed all point of the reviewer.
However, they seem to be incorrect in one of their argument and should re-check this issue again:
L96 “elevated by 152.3%” elevated TO instead of BY?
L147 similarly “To” instead of “by” 163% Please check throughout the text for the correct use of to xy% and by xy% (seems to be incorrect for values above 100%)
A: As authors know, both ‘by’ and ‘to’ are widely used in the other articles. In addition, this manuscript has been received the English editing service by the professional foreigners.
The authors’ argument is true, that both “by” and “to” are widely used in English, but they have a different meaning: an increase from 2 to 3 would be an increase BY 50% ([3-2]/2=0,5) as original value), but it would be an increase TO 150% (3/2=1,5). Therefore, it is very important to use the right one. To the reviewer, it does not always seem to be correct, mostly for increased to values above 100% is seems to be incorrectly chosen.
First example original Line 96 now Line 98:
“adiponectin levels in the LCP group were significantly elevated by 152.3%”
The figure 2g for adiponectin shows about 18 µg/ml for CON as control and about 25 µg/ml for the LCP bar, which would mean an increase TO 25/18 = 139% or an increase BY 39%. Given the data from the graph, it looks much more like an elevation TO 153% then an elevation BY – the LCP bar would need to bar 2.5 times larger which it isn’t.
I can understand that the professional foreigners would not pick up this problem as the English language itself is correct;, only if they would have calculated all the data, they would have noticed the problem – something language editing service would not perform. Please check all % data in the paper.
Author Response
Dear Editor and reviewer
We have revised our manuscript very carefully based on reviewer’s comments. We are grateful to the critical comments that have helped us to improve the manuscript considerably. The revised parts are highlighted using a red font in the manuscript
A: As the editor and reviewer commented, we check all % data and revised the manuscript (line 98, 167, 190-192).
We hope that the revised manuscript is now acceptable for publication in the ‘Biomedicines’. Thank you very much for your consideration.

Reviewer 3 Report
The authors have sufficiently addressed the major concerns related to the primary manuscript submission, however, please indicate the size of the scale bars (microns or micrometers) that were added to histology images.
Author Response
Dear Editor and reviewer
We have revised our manuscript very carefully based on reviewer’s comments. We are grateful to the critical comments that have helped us to improve the manuscript considerably. The revised parts are highlighted using a red font in the manuscript
A: As the editor and reviewer commented, we added the size of scale bar in the legend of Figure 2 and 3.
We hope that the revised manuscript is now acceptable for publication in the ‘Biomedicines’. Thank you very much for your consideration.
